# Children of Smoking and Non-Smoking Households’ Perceptions of Physical Activity, Cardiorespiratory Fitness, and Exercise

**DOI:** 10.3390/children8070552

**Published:** 2021-06-26

**Authors:** Melissa Parnell, Ivan Gee, Lawrence Foweather, Greg Whyte, Zoe Knowles

**Affiliations:** 1Public Health Institute, Liverpool John Moores University, Liverpool L2 2QP, UK; m.parnell@2016.ljmu.ac.uk (M.P.); I.L.Gee@ljmu.ac.uk (I.G.); 2Research Institute for Sport and Exercise Sciences, Liverpool John Moores University, Liverpool L3 3AF, UK; G.Whyte@ljmu.ac.uk (G.W.); Z.R.Knowles@ljmu.ac.uk (Z.K.)

**Keywords:** physical activity, fitness, children, secondhand smoke, smoking, qualitative, low socioeconomic status

## Abstract

Previous research has shown secondhand tobacco smoke to be detrimental to children’s health. This qualitative study aimed to explore children from low socioeconomic status (SES) families and their reasons for being physically active, attitudes towards physical activity (PA), fitness and exercise, perceived barriers and facilitators to PA, self-perceptions of fitness and physical ability, and how these differ for children from smoking and non-smoking households. A total of 38 children (9–11 years; 50% female; 42% smoking households) from the deprived areas of North West England participated in focus groups (*n* = 8), which were analysed by utilizing thematic analysis. The findings support hypothesised mediators of PA in children including self-efficacy, enjoyment, perceived benefit, and social support. Fewer than a quarter of all children were aware of the PA guidelines with varying explanations, while the majority of children perceived their own fitness to be high. Variances also emerged between important barriers (e.g., sedentary behaviour and environmental factors) and facilitators (e.g., psychological factors and PA opportunity) for children from smoking and non-smoking households. This unique study provided a voice to children from low SES and smoking households and these child perspectives could be used to create relevant and effective strategies for interventions to improve PA, fitness, and health.

## 1. Introduction

Cardiorespiratory fitness (CRF) is a health-related component of physical fitness defined as the ability of the circulatory, respiratory, and muscular systems to supply oxygen during sustained physical activity [1]. CRF is an established indicator for health in children and adolescents [2]. Furthermore, CRF during childhood and adolescence is positively associated with cardiovascular health in later life [3,4,5]. Yet, CRF among children and adolescents from high and upper-middle income countries has substantially declined since the 1980s, with stabilisation in the trend since 2000 [6]. CRF in children has also decreased over time in the North West of England, with 35.8% of boys and 59.7% of girls classified as unfit according to established CRF thresholds [7], thus highlighting the importance of early intervention efforts to promote CRF in this region. Physical activity (PA), in particular moderate-to-vigorous intensity PA (MVPA), is strongly associated with CRF [8,9] and low PA in childhood is predictive of low PA in adulthood [10,11]. Yet in 2019, only 51% of boys and 43% of girls met the United Kingdom (UK) PA guidelines [12], which states that children and youth aged 5–18 years should achieve at least an average of 60 min of MVPA daily [13]. Lower socioeconomic status (SES) has been shown to be associated with lower levels of PA [14,15] and physical fitness, including CRF [16,17] in youth. Health interventions aimed at improving CRF in children should therefore consider MVPA in addition to other factors associated with health inequalities such as SES, diet, and tobacco smoke exposure.

Deeper understanding of low SES children’s perceptions, barriers, and facilitators of CRF and PA could be helpful in designing intervention and policy strategies. The literature regarding barriers and facilitators to children’s participation in PA [18,19,20] has highlighted the influence of multiple socio-ecological factors [21]. Psychosocial factors include perceived the availability of time, interest and motivation [19], parental support, and safety concerns; environmental factors include the availability of outdoor space and perceived safety [22,23]. Sedentary behaviour, including screen time, has been found to be a significant barrier to PA [20], with many children spending more time watching television with family members than engaging in PA with them [24]. Low SES has also been shown to be associated with lower levels of PA, but this association may be due in part to the comparatively hazardous neighbourhood environments [25] or home environments which have fewer opportunities for PA [22,24]. As far as CRF is concerned, genetics, sex, age, and maturity are well documented non-modifiable determinants of CRF [26]. PA is an established modifiable determinant of CRF [2,27], along with diet and body mass [1,25], while SES has been shown to influence CRF independently of PA [28]. Wider individual, social, and environmental determinants of CRF, particularly in children, are unclear and this information could support the development of interventions to improve CRF. Although low-SES populations are increasingly targeted for interventional research [29,30], children from low-SES communities may face different barriers and facilitators to PA and CRF compared to their high-SES counterparts of which there are less research.

Article 8 of the World Health Organisation (WHO) Framework Convention on Tobacco Control (FCTC) states that individuals have a right to a tobacco smoke-free environment [31]. Since the WHO recommended compliance with Article 8 of FCTC [32], smoke-free policies have been increasingly adopted all over the world [33] and there is consistent evidence that national smoking bans have improved cardiovascular health outcomes and reduced mortality for associated smoking-related diseases [34]. The UK has been a strong adopter of the FCTC and, in England, children’s exposure to second-hand smoke has fortunately declined by 79% since 1998 due to the emerging social norm of smoke-free homes [35]. The smoking ban, which came into effect July 2007, as a result of the Health Act 2006 made it illegal to smoke tobacco in enclosed places in England, with similar bans already introduced in Scotland, Wales, and Ireland earlier. Other tobacco control measures such as standardised packaging [36], raised taxes to increase the price of tobacco [37] marketing restrictions, graphic health warnings, and cessation treatment policies [38] have also contributed to the decline in smoking prevalence in the UK. Additionally, in 2015, England became one of the first nations to implement a law prohibiting smoking in private vehicles with children present (The Smoke-free (Private Vehicles) Regulations, 2015) [39,40]. Despite these tobacco control measures, however, smoking is still permitted in private residences and two main determinants of children’s SHS exposure in England have been reported to be smoking by parents or caregivers and whether smoking occurs in the home [41].

Children from low-SES households are more likely to be exposed to secondhand smoke (SHS) [42,43] and consequently more likely to suffer the detrimental impacts of SHS exposure [44]. Exposed children are at increased risk of chronic airway inflammation, lung function defects [45], severe asthma attacks, respiratory infections, ear infections, sudden infant death syndrome [46], and increased risk of hospitalisation in asthmatics [47]. The negative health effects of SHS exposure have physiological implications for PA and CRF, with SHS exposure associated with reduced exercise performance [48,49,50] and increased blood pressure in exercising adolescents [51,52]. Children from smoking households could therefore be at greater risk of low fitness and the associated health consequences. SHS exposure may also have indirect effects on children’s engagement in PA and, therefore, CRF through influencing their perceptions of PA, CRF, and exercise (i.e., attitudes, beliefs, feelings, and emotions). No research has yet explored how children’s perspectives surrounding PA, CRF, and exercise compare relative to children of smoking and non-smoking households. A deeper understanding of the barriers to PA and CRF that low SES children face and whether household smoking status is a significant factor, will allow better informed health intervention and health promotion strategies for this population.

When it comes to understanding the experiences and views of children, children are the experts with their own unique perspective of the world [53]. Qualitative methods can assist in capturing children’s understanding and perceptions of physical activity and CRF and whether these experiences and perceptions differ by household smoking status. Focus groups involving children have been previously used to explore children’s perspectives and attitudes towards PA [54,55] and children’s thoughts and feelings when they are exposed to SHS [56,57]. Supplementing focus groups with activities such as the write, draw, show, and tell (WDST) method [54] can keep children interested and engaged and can further allow children to express their ideas in a manner such that researchers can assess children’s meanings [58]. This study therefore aimed to use creative qualitative methodologies to explore the perceptions of children (9–11 years) from smoking and non-smoking homes surrounding cardiorespiratory fitness and physical activity. The study sought to address the following research questions (RQs).

(RQ1) What are the reasons that children from smoking and non-smoking households have for being physically active?

(RQ2) What are the attitudes that children from smoking and non-smoking households have towards physical activity, exercise, and fitness? 

(RQ3) What are the perceived barriers and facilitators to a child’s ability to be physically active and does this differ for children from smoking and non-smoking homes? 

(RQ4) What are the children’s perceptions of their own fitness and physical ability and does this differ for children from smoking and non-smoking homes?

## 2. Materials and Methods

### 2.1. Study Design

This research was granted ethical approval by the University Research Ethics Committee (Ref: 16/PBH/001) and follows the Consolidated Criteria for Reporting Qualitative studies checklist [59]. The study was approached with a humanistic philosophy by acknowledging children as experts that possess their own unique perspective of the world [53]. This unique perspective was explored using creative qualitative methodologies [54] and gave a voice to children from both smoking and non-smoking households. Participants were drawn from a concurrent quantitative investigation into the associations between smoking exposure, CRF, and child health and was conducted as part of a wider programme of research [48]. Data collection began in September 2017 and ended in February 2019, with schools participating at different timepoints throughout the year and determined according to convenience relative to the schools.

### 2.2. Participant Selection and Setting

Participants were targeted as being aged 9–11 years old and in year 5 or 6 at a state-funded primary school within two metropolitan boroughs in North West England. This age group was targeted since evidence from North West England has reported low fitness among primary school children [7] and therefore this study sought to examine whether smoking exposure was a factor. Children’s thoughts and feelings may begin to have stronger influences on activity and exercise in older childhood and adolescence [60] and the examination of this age group is therefore an important form of prevention perspective. One-hundred and forty-seven schools were approached as convenience samples and four schools agreed to take part in the research (2.7% response rate from schools), with all participating schools falling within the two most deprived deciles within the English Indices of Multiple Deprivation (EIMD) based on the school post code [61]. The low school participation rate is likely due to the controversial nature of the research and academic timetable pressures, with one school noting that the research was too contentious for them to participate and others noting that they were too busy.

After receiving written informed gatekeeper consent from headteachers, presentations were given at the participating schools to provide information to the children about the research and to invite children to take part. Information packs, including parental questionnaires, parental consent forms, and child assent forms were given to children to take home to parents and guardians. One-hundred and five children returned parental consent and child assent and those were eligible to take part in the wider programme of research (26.5% response rate from invited families). 

Questionnaires intended to determine self-reported household smoking status, using items from the Global Adult Tobacco Survey (GATS) by the Global Adult Tobacco Survey Collaborative Group [62], were sent home to be completed by a consenting parent or guardian. Questions determined the number of tobacco smokers living in the home, as well as questions inquiring which rooms smoking occurred and/or was permitted and how many cigarettes were smoked each day per person. Space was provided for participants to include information regarding smoking habits for up to four members of the household, with more space available upon request. Similar information was collected for e-cigarette use. Participants were classified into ‘non-smoking household’ or ‘smoking household’ according to whether a household member reported they smoked cigarettes or not, regardless of where smoking was permitted. Households with an adult that used e-cigarettes were classified as non-smoking, as earlier quantitative aspects of the research found no significant differences in fitness and health outcomes between children from non-smoking and ‘vaping’ homes. The quantitative aspect of the larger research project measured exhaled carbon monoxide (eCO) measurements, but no significant differences were observed between eCO concentrations between the groups. In the present study, whilst eCO concentrations were elevated in for children from smoking homes (1.17 ppm) compared to children from non-smoking homes (1.02 ppm), this was not statistically significant (*p* = 0.721).

Participant (adult and child) demographic information and child medical history were obtained via the parental questionnaire. Household deprivation was assessed via the EIMD using participant home postcode and the Ministry of Housing, Communities, and Local Government postcode lookup tool [61].

At each participating school (*n* = 4), two focus groups were held: one with children from non-smoking households; and another with children from smoking households. Parental surveys used in the initial phase of the research [48] identified children from smoking and non-smoking households and were used to inform focus group membership. Focus group participants from each smoking exposure group were selected by stratified sampling, with the number of boys and girls controlled for to allow even representation. All eight focus groups involved the recommended group size of 4–6 participants [63,64]. For some groups, there were not enough children identified from smoking households to meet the recommended group size. Therefore, in order to avoid excluding children from smoking households due to low numbers, children from non-smoking households were also invited to join in the focus group.

Focus group membership is outlined in Table 1. A sub-sample of 38 children, including 19 boys and 19 girls, participated in the focus groups. Forty participants were selected but one boy and one girl were absent at the time of data collection. The majority of participants were from non-smoking households, including 11 boys and 11 girls, with 16 children from smoking households, including 8 boys and 8 girls. The average age of the focus group participants was 10.2 years, with white British children making up 65.8%, Black British 10.5%, 7.9% white-other (including Polish and Portuguese), and 15.8% of participants were of other ethnicities. The majority of the participants’ homes (79%) were amongst the most deprived two deciles for neighbourhood deprivation in England [61]. Other than household smoking status and deprivation level, the focus groups were fairly homogeneous with a similar prevalence of asthma and did not have any other medical conditions that may have influenced responses.

### 2.3. Focus Groups

Eight semi-structured, mixed gender, and child-centred focus groups were facilitated by the first author (female) following training. Focus groups took place in a familiar school setting (in a classroom or staffroom at the participants’ school), during school time, and in a place where participants could be overseen but not overheard to comply with safeguarding procedures [56].

The four principal research questions informed the production of an age-appropriate focus group guide which encouraged children to consider their own thoughts, opinions, and beliefs (available upon request). Focus group questions were reviewed by a Health and Care Professions Registered Psychologist for age appropriateness with ordering and flow designed to facilitate interaction between children. Focus groups exploring children’s perspectives should be small in number and interactive to maintain a high level of interest [56]. The focus group design was therefore influenced by the recently established write, draw, show, and tell (WDST) method, which is an inclusive, interactive, and child-centred methodology [42]. Although drawing was not employed as a method in the current study, the visual methods such as ‘write’ and ‘show’ were used in combination with verbal articulation from the children. Most questions permitted thinking time, which allowed children to consider their own thoughts and opinions before sharing with the group. Interactive questions, for example with the use of sticky notes, offered an opportunity for children who were less comfortable sharing their thoughts verbally to contribute to the discussion. Further detail of the focus group activities is provided as Appendix A.

All focus groups were recorded by Dictaphone and field notes were not taken due to the level of interaction and facilitation required throughout the sessions. Verbal consent was sought from each child before the focus group commenced, following an explanation to the participating children from the facilitator. The children were told there were no right or wrong answers and that the focus group intended for them to share their thoughts and opinions, but they did not have to answer if they did not want to. The focus groups started with introductions, basic group expectations (e.g., ‘Please do not try to talk over each other’), and an icebreaker was used to allow the children to practice speaking freely in the group. Participants were provided with the opportunity to provide any further thoughts and opinions on the focus group topics at the end of each focus group.

### 2.4. Data Analysis

Focus groups lasted an average time of 36 min (range 29–44 min) and were audio recorded and transcribed verbatim, which resulted in 118 pages of Arial size 12 font, double spaced, and raw transcription data. The first author was the sole coder, but generated themes were discussed and refined with the wider research team. Participants did not provide feedback on the findings but had been provided with the opportunity to provide any further thoughts and opinions at the end of the focus group. Quotations are presented verbatim, with reference to the participant number, sex, school, and household smoking status (more detail below).

### 2.5. Migration of Data

In two focus groups, children from non-smoking households were included in the ‘smoking household’ focus groups (FG2 and FG4) for reasons described above. Where possible, the data obtained from the focus group attributable to these children, e.g., quotes, pictures, sticky note activity data, and agreements/disagreements was migrated into the ‘non-smoking household’ dataset. This was made possible due to the lack of discussion between the children. In the focus groups, children tended to answer the questions in relation to themselves, with little comparison and contrast between other group members’ answers. Data migration was therefore practical and did not impact the remaining set or the set to which it was added. However, where discussion, agreement or disagreement did occur, a larger portion of the data was migrated in order to refrain from losing the context of the discussion and to ensure reverse tracking within analysis procedures.

### 2.6. Thematic Analysis

Thematic analysis was employed to analyse the data, the process of which consisted of six stages: 1—familiarisation with the data; 2—generation of codes; 3—generation of initial themes; 4—reviewing themes; 5—defining and naming themes; 6—synthesis of the report [65]. Verbatim transcripts were read and re-read to allow familiarisation of the data and then imported into the QSR NVivo 10 software package. Items of interest and initial thoughts and ideas were noted during the familiarisation phase. Codes were generated inclusively, comprehensively, and systematically and the codes captured data that were related to the research questions. Themes were generated as an active process by organising smaller amounts of data associated with codes into larger clusters of data with similar codes to produce themes. Thematic maps and tables were used to visualise and consider the relationships between themes and review potential themes. Themes had to have meaningful data in support and those that did not have enough data were discarded. Stage five of the thematic analysis involved defining and naming themes. The themes were described in their relation to the overall ‘story’ and with respect to the answers for the research questions. Finally, the report was produced as an analytic commentary, using quotes and extracts from the data to demonstrate the themes generated in relation to the research questions. Quotations are labelled by the participant: boy (B); girl (G); ID number; school (A,B,C, and D); household smoking status, smoking (S); and non-smoking status (NS). For example, B6B/NS, would stand for boy 6 from school B and a non-smoking household.

### 2.7. Pen Profiles

A pen-profiling approach, which is increasingly used to report and support creative methodologies [66], was used to represent thematic analysis outcomes. Pen profiles are a method of data presentation that incorporate numerical data (number of responses), thematic outcomes, and verbatim quotes in a clear and succinct manner. Pen profiles are considered appropriate for representing analysis outcomes from large datasets via a diagram of composite key developed themes [63]. In order to expand the pen profiles, verbatim quotations were used directly from the transcripts. This technique presents findings in a manner that is accessible to researchers who have an affinity for both qualitative and quantitative backgrounds [66]. Percentages within the pen profiles represent the proportion of each group that contributed to the theme for children from smoking and non-smoking homes, and percentages within the text represent the whole sample, unless stated otherwise.

## 3. Results

### 3.1. What Is Physical Activity?

Words for describing the physical activity used by the children in the icebreaker activity could be categorised into types of physical activities such as team sports, organised activities, and solo activities (*n* = 72); words for describing what physical activity is (*n* = 32) and how it made the children feel is (*n* = 8). Physical activity was most commonly associated with sports (*n* = 15), football (*n* = 12), running (*n* = 10), and swimming (*n* = 8). When assessing smoking exposure groups separately, children from non-smoking households more frequently used descriptors such as ‘sports’ (*n* = 11), ‘health’ (*n* = 6), and ‘fun’ (*n* = 7), while children from smoking households were more likely to give examples of physical activities such as ‘football’ (*n* = 6) and less likely to describe physical activity as ‘fun’ (*n* = 1). Although not constituting a theme, one participant from a smoking home described physical activity as ‘tiring’, which was the only negatively perceived description of PA by any of the participants.

### 3.2. RQ1. What Are the Reasons Children from Smoking and Non-Smoking Households Have for Being Physically Active?

Football was the most common favourite physical activity (*n* = 11, 29%), followed by swimming (*n* = 5, 13%), and dance (*n* = 5, 13%). Children from non-smoking homes typically chose more metabolically demanding activities as their favourite physical activity, for example, martial arts and cycling, whereas children from smoking households often favoured less metabolically demanding activities, such as walking and darts, although both groups frequently favoured football and swimming.

A pen profile representing children’s reasons for being physically active is presented in Figure 1. Four major themes were generated in response to the first research question, including positive feeling (*n* = 19, 50%), perceived benefit (*n* = 15, 39%), perceived competence (*n* = 5, 13%), and social influence (*n* = 7, 18%).

The positive feeling theme consisted of three sub-themes including fun (*n* = 8, 21%), enjoyment (*n* = 6, 16%), and feels good (*n* = 5, 13%, NS only). Fun and enjoyment were common reasons associated with participating in physical activity among both groups. One child noted enjoying the feeling of competition he experienced from PA, “I just like being competitive….and in dodgeball you can throw balls at people and just whack them!” (B19D/S).

Perceived competence (*n* = 5, 13%) was a theme identified from the responses from children from non-smoking homes only. Reasons for being physically active provided by non-smoking children, which related to their perceived competence, included prior experience (*n* = 2, 5%) and ability (*n* = 3, 8%).

A third theme, social influence (*n* = 7, 18%), made up of friends (*n* = 6, 16%) and family (*n* = 1, 3%) was reflected in the responses of children from both smoking (*n* = 3) and non-smoking households (*n* = 3). One participant stated that playing Xbox was his favourite (physical) activity for reasons related to playing with friends: “Playing on my Xbox. Because I can play with my friends. And I get to play *Fortnite*, and it warms my thumbs up.” (B12C/S). When discussing whether playing Xbox was a physical activity or not, a minority of children believed it could be classed as PA (*n* = 2, 5%) as it involves ‘moving your thumbs’ (B12C/S).

Children reported reasons associated with a perceived benefit to being physically active (*n* = 15, 39%), with children from non-smoking homes referring to benefits of PA more often (*n* = 12) than children from smoking households (*n* = 3). For example, exercise was a reason often provided by children from non-smoking households (*n* = 7), “Football, because you need some exercise.” (B3A/NS). Health reasons, for example, “Football, because it’s healthy and fun” (B2A/NS), were frequently reported by children from non-smoking homes (*n* = 4) and by one child from a smoking home. Children also reported the benefit of learning new skills and techniques as a reason for taking part in PA (*n* = 3, 8%).

### 3.3. RQ2. What Are the Attitudes of Children from Smoking and Non-Smoking Households towards Physical Activity, Exercise, and Fitness?

#### 3.3.1. Perceptions of Physical Activity Guidelines

Children were asked how much physical activity they believe they should be involved in per day in order to assess their current understanding of the physical activity guidelines. Overall, the most common answer was 60 min per day (*n* = 8, 21%), followed by 90 min (*n* = 7, 18%) and 120 min (*n* = 7, 18%) per day. Children from non-smoking households frequently stated that children should do 60 min of PA per day (*n* = 6, 27% of children from non-smoking homes), whereas children from smoking households most frequently stated that children should do 90 min of PA per day (*n* = 5, 28% of children from smoking homes).

#### 3.3.2. Importance of Fitness

All participating children believed that it is important to be physically fit and statements from the participants indicate that children conceptualise fitness more widely than only cardiorespiratory fitness. For example, one child noted that a ‘fast’ child must exercise a lot, “If someone’s faster than me, then I think that they must be doing a lot of exercise.” (B1A/NS). A higher order theme generated for why children believed physical fitness to be important was capability (*n* = 24, 63%), which was split into three sub-themes (Figure 2): physical activity and sport performance (*n* = 13, 34%), physiological aspects of ability (*n* = 6, 16%), and future capability (*n* = 5, 13%). Children from non-smoking homes valued fitness in terms of performance (*n* = 10, 45% of children from non-smoking homes) in PA such as sport and games, for example “Like you play a game of tag or something, and someone’s tagged you and you’re on, you need to be fit to try and get them.” (B2A/NS). Children from smoking homes more often talked about the physiological impacts of fitness (*n* = 4, 25% of children from smoking homes): “…because if you don’t keep physically fit, you’re just going to run out of breath all the time when you’re walking somewhere or down somewhere at the park…” (B14C/S). Fitness was believed to be important for the future by children from non-smoking homes (*n* = 5, 23% of children from non-smoking homes), “Well, it’ll [fitness] help you in your future” (B10B/NS). However, children from smoking homes did not discuss fitness being important for the future. Although not constituting a major theme, self-esteem was discussed by children from both exposure groups (*n* = 3, 8%). Children reported fitness was important because “…you get more confidence from it [fitness]” (G18D/S) and “It’s [fitness] important to me because you could get bullied and stuff because you’re not fit…” (B7B/NS).

The theme of health benefits of fitness constituted two minor themes including general health (*n* = 7, 18%) and weight status (*n* = 5, 13%). Children from non-smoking homes were more likely to report reasons surrounding the health benefits (*n* = 7): “Because it’s [fitness] good for your body and your bones and stuff” (G8B/NS), whereas children from smoking homes did not discuss the health benefits directly. Fitness was important to children from smoking and non-smoking homes for reasons linked to weight status (*n* = 5), with children from both groups relating fitness to fatness.

#### 3.3.3. Improving Fitness

The consensus from all participants was that children can improve their fitness, for example:

“I think you can always improve your fitness, because you can improve it by doing more workouts and stuff, but I think it will never be a ten [out of ten]. You can always improve it, because I think you can always get better.” (G1A/NS).

Children believed they could improve their fitness by increasing their level of PA (*n* = 17, 45%), which is a theme constituted of two sub-themes (Figure 3): exercise (*n* = 15, 39%) and sports (*n* = 2, 5%). In terms of intensity for improving or maintaining CRF, some children stated that they should build up the exercise intensity throughout the activity (*n* = 5, 13%). For example, “I think we [children] should start at quite light, and then like build up” (G3A/S). Some children believed they should work hard throughout (*n* = 4, 11%), “Hard [exercise intensity], so… like 100%” (G1A/NS) and some stated they should put ‘medium’ effort in (*n* = 3, 8%), whereas some children believed they should work as hard as they feel like at the time (*n* = 2, 5%), “I feel how much I want to do. If you don’t want to do that much that day, don’t do that much.” (B3A/NS). Increasing exercise frequency and intensity were the most common themes discussed by children from non-smoking homes (*n* = 10) and smoking homes (*n* = 5).

Getting outdoors to improve fitness was a major theme generated from the responses from both groups of children (*n* = 15, 39%). Children often commented on how going outdoors could improve their fitness. For example: “Going to the park, going out for walks, runs, going on my scooter” (B4A/S). When discussing going outdoors, children often expressed parental restriction due to safety concerns as a limiting factor, for example, “[Adults should] let you go outside all the time, even if it’s raining or anything” (G11C/NS). One participant spoke about how she was often grounded but could improve her fitness by going outside more.

A theme of significant for others (*n* = 7, 18%) was developed which included three sub-themes: parental support (*n* = 3, 8%), friends (*n* = 2, 5%), and dogs (*n* = 2, 5%). Both groups referred to the importance of friends in improving fitness. However, only children from smoking homes discussed dog ownership, whilst only children from non-smoking homes discussed parental support as a method that they could improve their fitness: “Because Mum and Dad can drive, they can take me out places where I can get fit” (B11C/NS).

A good diet (*n* = 5, 13%) was considered an important factor for improving fitness by children, with more children from non-smoking homes (*n* = 4) discussing diet than children from smoking homes (*n* = 1). One girl explained how, in order to improve her fitness, she might change her diet with the involvement of her parent, “I would say to my Mum, I’m not having any like carbs or junk for maybe two months or something” (G8B/NS).

A second minor theme generated by the responses of children was centred around the provision and availability of equipment, for example, the ownership of bicycles, scooters, trampolines, and treadmills. Some children stated that having a treadmill at the home allowed them to increase their fitness (*n* = 2, 5%), but one girl from a smoking household noted how no one in the house uses the treadmill, “But we’re getting rid of [the treadmill] soon. Only the dog uses it” (G4A/S).

### 3.4. RQ3. What Are the Barriers and Facilitators to a Child’s Ability to Be Physically Active and Does This Differ for Children from Smoking and Non-Smoking Homes?

#### 3.4.1. Barriers

Children identified a range of factors which limited their ability to be physically active (Figure 4). The majority of factors identified by children from both smoking and non-smoking homes were associated with sedentary behaviours (*n* = 29, 76%), including screen time (*n* = 16, 42%) and other general sedentary behaviours (*n* = 13, 34%). Screen time was described to be a major factor preventing children from being physically active, for example: “If my sister didn’t go to school, she would spend all day in bed, literally, watching YouTube” (B12C/S) and “I’m always on my laptop. That’s all I’m ever on at home” (B7B/NS).

A second higher order theme of resources was generated (*n* = 8, 21%), which was linked to two sub-themes which are money (*n* = 3, 8%) and time (*n* = 5, 13%). Children from non-smoking homes were especially concerned with the amount of free time they had available to be active, particularly due to commitments to organised activity clubs outside of school. Money was discussed as a limiting factor by children from both exposure groups in terms of requiring money to pay for various physical activities.

Psychological factors were a theme generated from the responses from both groups of children as a factor which limits a child’s ability to be physically active (*n* = 9, 24%). A negative psychological state, for example feeling lazy or tired, was believed by the children to be a limiting factor relative to their ability to be physically active as they were less motivated to do so.

Physiological factors (*n* = 9, 24%) were noted as barriers and consisted of two subthemes: dietary habits (*n* = 6, 16%) and health and injury (*n* = 3, 8%). The latter was discussed by both groups of children, whilst nutritional factors were only discussed by children from non-smoking homes. Environmental barriers (*n* = 7, 18%) to activity, consisting of school (*n* = 6, 16%) and transport (*n* = 1, 3%), were reported by both agroups and particularly with regard to the sedentary nature of school-work and homework.

#### 3.4.2. Facilitators

Children commonly discussed the physiological factors (*n* = 20, 53%) that facilitate their ability to be physically active (Figure 5). Three sub-themes made up the physiological factors theme: dietary habits (*n* = 10, 26%), health (*n* = 6, 16%), and sleep (*n* = 4, 11%). Children from non-smoking homes more frequently talked about health and diet compared to children from smoking homes who more often reported sleep as important factor facilitating their ability to be physically active.

A theme of significant others was generated (*n* = 18, 47%), which consisted of four sub-themes: friends (*n* = 6, 16%), adults (*n* = 5, 13%), siblings (n = 4, 11%) and dog ownership (*n* = 3, 8%). Friends was an important factor for PA facilitation for children from both smoking (*n* = 3) and non-smoking households (*n* = 3). Children from smoking households more frequently mentioned the influence of adults (*n* = 4). For example, ‘Well, my Mum helps me be active. Well, when I ask if I can play out, she’s like, “Just get out”…’ (G14C/S). Dog ownership was referred to as a factor facilitating PA with children from non-smoking households only (*n* = 3).

Opportunity for physical activity (*n* = 14, 37%) was a theme generated from responses of children from both smoking and non-smoking households, with participation in various clubs and different types of physical activities noted. Although it did not constitute a theme, one child noted that being active from a young age would encourage PA later in life: “Like if you be really active when you’re little, then you can grow up to be more active. Like you want to be active then.” (G6B/NS).

Psychological factors were also discussed positively in relation to factors which facilitate a child’s ability to be active (*n* = 9, 24%), with children from non-smoking homes (*n* = 6) and smoking households (*n* = 3) describing positive attitudes such as ‘determination and commitment’ (B3A/NS) and ‘when you’re energised.’ (G4A/S).

A theme of environment (*n* = 7, 18%) was generated which consisted of outdoors (*n* = 4, 11%) and transport (*n* = 3, 8%). Outdoors was a factor only discussed by children from non-smoking homes, e.g., “…like say if you were playing out or something. You could play tag and that’ll give you exercise and things.” (G7B/NS). Transport was discussed by both children from smoking homes (*n* = 3, 8%).

#### 3.4.3. How Do Adults Limit or Facilitate Children’s PA according to Children from Smoking and Non-Smoking Households?

Children from smoking (*n* = 14) and non-smoking households (*n* = 10) commonly identified parents as positive influences on their ability to be physically active (*n* = 24, 63%) (Figure 6). Less frequently, parents were identified as negative influences (*n* = 6, 16%) for children from smoking (*n* = 2, 13% of children from smoking homes) and non-smoking homes (*n* = 4, 18% of children from non-smoking homes). Other family members including siblings were identified as positive influences on PA by children (*n* = 4, 11%) of non-smoking (*n* = 3) and smoking homes (*n* = 1). Coaches and teachers were identified as positive influences by children from non-smoking households only (*n* = 11, 50% of children from NSH), whereas friends of family were identified as positive influences by children from smoking homes only (*n* = 2, 13% of children from smoking homes).

Four higher order themes were generated in response to the discussion about how adults influence children’s ability to be physically active including provision (*n* = 21, 55%), instruction (*n* = 8, 21%), encouragement (*n* = 7, 18%), and restriction (*n* = 4, 11%). Provision was split into two sub-themes, which are logistical and financial support (*n* = 10, 26%), and these describe the provision of financial and logistical support for participation in sports clubs, training, and organised activities; and provision of opportunities for PA (*n* = 11, 29%), which describes physical activities which are not part of sports clubs or regular training. Children from smoking (*n* = 3, 19%) and non-smoking homes (*n* = 7, 32%) often commented on how adults facilitate organised PAs through logistical and financial means, for example, “[Adults] take you to football training.” (B2A/NS). Children from smoking homes discussed provision of opportunities for PA (*n* = 7, 44%) more frequently than children from non-smoking homes (*n* = 3, 18%): “My Mum and Dad normally walk me round the block and all that, and then sometimes I go on a bike ride with my Dad.” (G4A/S).

Children from non-smoking homes described how encouragement from adults helped them to be physically active (*n* = 7, 32%), for example, “My bother goes to 5Fit and makes me want to go” (B7B/NS), whereas children from smoking homes did not discuss encouragement. Rather, children from smoking homes discussed instruction from adults (*n* = 3, 19% of children from smoking homes), as did children from non-smoking homes (*n* = 5, 23% of children from non-smoking homes). Children described how parents (in both smoking and non-smoking homes) will give instructions to be more physically active, for example, “My Mum will tell me to go outside and have a play outside instead of sitting in.” (G4A/S) and “My mum tells me to go on a run with my sister, or if she doesn’t do that, she tells me to take the dogs.” (G15D/NS).

The theme of rules and restrictions was comprised of two sub-themes: grounding (*n* = 3, 11%) and safety (*n* = 1, 3%). Children from both groups reported that grounding as a punishment limited their ability to be physically active. One child from a smoking home also reported that parental concerns for safety prevented him from being physically active, whilst another child reported that they were not always able to go places to be physically active for logistical reasons, “I have to stay at home because there’s not enough room in the car.” (G13C/S).

### 3.5. RQ4. What Are Children’s Perceptions of Their Own Fitness and Does This Differ for Children from Smoking and Non-Smoking Homes?

Children were asked, “How physically fit do you feel, on a scale of 1–10? With 1 being not very fit at all, 10 being the fittest you could be.” The median self-perceived fitness score given by children from non-smoking homes was 8.0 (range 2–9) and 8.0 (range 1–10) for children from smoking homes (Mann–Whitney U test, *p* = 0.925). While only 6% (*n* = 1) of children from non-smoking homes rated their own fitness at the maximum level (10 out of 10), 21% (*n* = 3) of children from smoking homes rated their own fitness at the maximum level. Two children from smoking homes rated their own fitness as 1, whereas the lowest score provided by the children from non-smoking homes was 2.

Children were provided with five photographs (available within the Appendix A). Participants were asked to put the photographs on the Pictorial Children’s Effort Rating Table (PCERT) scale [67], which they were familiar with from the laboratory-based aspect of the wider research project [48]. All but two (95%) of the participants rated walking as the easiest activity. Differences were observed between household smoking status groups as well as sex differences between how difficult the children rated the remaining activities. Overall, at least half of boys (*n* = 4, 50%) and the majority girls (*n* = 5, 63%) from smoking homes rated running as the hardest activity with descriptions such as “Running, it’s kind of hard because it tires me out” (B4A/S) and “I just don’t like running” (G5A/S). Boys and girls from non-smoking homes did not rate running as the hardest, but most commonly rated gymnastics (*n* = 7, 64%) and monkey bars (*n* = 5, 45%), respectively, as the hardest activities. Table 2 summarises the consensus from boys and girls from smoking and non-smoking homes as to the difficulty of each physical activity.

Once the children had arranged the pictures onto the PCERT scale, they were asked to describe and explain their choices. Example responses from one girl (smoking household) and one boy (non-smoking household) are listed below.

Participant G4A/S explained her choices in the following:

“Walking a one, swimming a two, running a five or a six, gymnastics a three, and the park a four. Walking’s easy because it’s everyday stuff. Swimming, I’ve been swimming since I was three and a half, so it’s kind of in my blood. Gymnastics, I’ve done that for a few years, so it’s like easy. The park, I go to the park all the time with my little brother and my Mum, my Dad and my dog. And running, I just don’t like running.” (G4A/S).

Participant B15D/NS explained his choices in the following:

“Walking easy, swimming, I find that just easy. Monkey bars are easy as well. Running is like easy at the start, then at the end gets harder. Crab, I can’t do that at all.” (B15D/NS).

When asked at which intensity on the PCERT scale the children would prefer to work at during physical activity, most children expressed that they prefer to work hard. The median preferred an intensity of 7 (range 4–10) and 10 (range 3–10) for children from non-smoking homes and smoking homes, respectively. Four children from smoking homes said they would prefer to work at an intensity of 10 out of 10, compared to only one participant from a non-smoking home. Many children commented that they would prefer to work at a range of intensities, “I would prefer to go there [10 out of 10] until I’m all tired out, and then I can just go down to one.” (B13C/S). One participant provided a reason for her choice of 9 out of 10: ‘I would say a nine because if it’s a ten, it [PCERT scale] says “so hard you’re going to stop”. You don’t want to stop, because then you don’t do nothing. But nine’s really, really, hard, so you’re working as hard as you can.’ (G3A/NS).

## 4. Discussion

This study aimed to explore attitudes, thoughts, beliefs, and perceptions surrounding physical activity and fitness of low SES children from smoking and non-smoking homes using interactive qualitative methodologies. The results demonstrate similarities and differences for children from smoking and non-smoking homes. Children noted that taking part in PA for reasons linked to positive feelings, social influence, and perceived benefit. Children from non-smoking households also noted that they took part in PA for reasons linked to perceived competence. Fitness was important to children from non-smoking households for health, performance, and future benefit, whereas children from smoking households believed fitness was important to them to avoid negative physiological consequences. Children believed more physical activity, significant others, the outdoors, active equipment, and a good diet could assist them in improving their fitness. The perceived barriers and facilitators to PA were centred around psychological factors, physiological factors, significant others, the environment, resources, sedentary behaviour, and opportunity for PA. The majority of children perceived their CRF to be higher than their actual CRF level. Variances were observed for the ranking of physical activities by difficulty between boys and girls, and exposure group. A handful of themes, including significant others, opportunity for PA, health, and the outdoors, were found to be especially significant to participants and the overlap of these themes was apparent across the research questions.

### 4.1. What Is Physical Activity?

When prompted to describe physical activity in three words, the most common words used by both groups of children included sports, football, running, and swimming. This finding may reflect the sports of the UK national curriculum for key stage 2 [68], with 68% of children age 7–11 years taking part in team sports, 46% taking part in running, and 31% taking part in swimming activities [12]. Whilst children from smoking households more often used words to describe what PA is (e.g., ‘fun’ and ‘sports’), children from non-smoking households used words to describe PA in terms of positive associations such as ‘fun’ and ‘healthy’. Associating PA with fun is important as ‘fun’ was reported as a major predictor of participation in PA [69], while enjoyment of PA at age 10 is associated with PA in adulthood [70]. The one negative word used to describe PA, ‘tiring’, was provided by a participant from a smoking household.

### 4.2. RQ1. What Are the Reasons Children from Smoking and Non-Smoking Households’ Have for Being Physically Active?

The findings support the main hypothesised mediators of physical activity in children: self-efficacy, enjoyment, perceived benefits [71], and social support [72]. Findings align with previous research identifying the top reasons why children found physical activity ‘fun’ as being: skilled and competent in PA; active with family members; learning new skills and knowledge; feelings experienced during movement; competition and winning [73]. Although Hopple [73] specifically explored why children find PA fun, whereas the current study examined reasons children take part, the same factors of feelings, social influence, perceived competence, and perceived benefit appear to be important factors in children’s PA. Reviews [74,75] have identified major correlates of PA in children including perceived competence, sensation seeking, and previous PA. These mirror themes of positive feeling, perceived competence, and the subtheme of prior experience from the current study.

Children from both smoking and non-smoking homes identified reasons for participation related to fun and enjoyment, but only children from non-smoking homes mentioned ‘feeling good’. For example, participants from non-smoking homes described positive feelings they get from swimming “…you feel good after you’ve been swimming” (B7B/NS) and cycling “I like the feeling of being able to go really fast really easily” (B17D/NS). Social Cognitive Theory [76] identifies cognitive (personal), behavioural, and environmental factors that influence behaviours. Outcome expectations are personal factors that relate to behaviour and when outcome expectations are positive, there is greater chance of engagement with the behaviour [77]. In the current study, positive outcome expectations were observed for both groups of children, although more frequently for children in the non-smoking group. Heitzler [78] found positive outcome expectations or beliefs about the benefits of PA to be related to children’s participation in PA. O’Dea [79] used focus groups with similar aged children, where participating children also highlighted enhancement of physical sensation as a benefit of PA. The results of the pilot study from the quantitative phase of the larger research project found participants from smoking households to have lower CRF levels than the children from non-smoking households [48], which may be reflected in their responses. In later focus group questions, children from smoking homes indicated that they find vigorous PA more difficult than their non-smoking household counterparts, which could explain why they do not refer to feeling good physically during their chosen PA. It is interesting that children from smoking homes did not discuss feeling good physically as a reason for participation, despite reflecting positively on their chosen favourite physical activities. This original finding warrants further research to explore any differences in how children from smoking and non-smoking homes feel when taking part in PA.

Children also reported taking part in PA for reasons linked to autonomous forms of extrinsic motivation through perceived benefits such as activities, including health and exercise, and to learn skills and techniques. Overall, these findings align with O’Dea [79], who found children’s perceived benefits of PA to include psychological status, physical sensation, sports performance, and social benefits. However, children from non-smoking households were more likely to report perceived benefits, such as exercise and health, than children from smoking homes: “Football, because it’s healthy.” (B2A/NS). Non-smoking adults are more likely to be physically active than smoking adults [80] and children from non-smoking homes may be echoing parents’ opinions when they state that they take part in PA because it is ‘exercise’ and ‘healthy’. This may indicate that participants from non-smoking households have greater health literacy [81] and physical literacy [82], demonstrating greater knowledge and understanding of the benefits of PA. Protheroe [83] showed that SES is associated with health literacy. Children whose parents have high educational background are more knowledgeable about health topics [84] and children from high and medium-SES perceive PA participation to be of greater importance [85]. Further research could explore the level of understanding regarding the benefits of PA and CRF for children from smoking and non-smoking households. Such information could inform interventions centred around health education, which could be tailored to low-income and/or smoking families.

A theme of competence, which consisted of ability and prior experience, was developed from the responses of children from non-smoking households only. Competence motivation theory states that individuals are driven to engage in activities to demonstrate their skills and high perceptions of competence results in increased competence motivation [86]. Reasons for taking part in PA relating to competence, e.g., “because I’m good at it”, were common for children from non-smoking homes. Although self-perceived competence is discussed further below, it is worth noting that De Meester [87] showed perceived motor competence to be associated with higher levels of PA regardless of actual motor competence. Parry [60] used longitudinal data to show that perceived ability at age 10 was associated with sport motivation at age 16 and that perceived ability is a crucial mediator of the relationship between participation and enjoyment. Perceived competence was not a reason provided by children from smoking households, which suggests other factors are more important drivers for PA in this group. Welk and Schaben [88] showed that when given similar opportunities to be active, some children will seek out different methods to be active whereas others choose to be less active; this is a finding thought to be mediated through perceived athletic competence. Self-perception of fitness is explored below but more detailed research exploring children from smoking households’ self-perceptions of motor competence could determine whether perceived competence is lower in this group, or less important than other drivers of PA.

### 4.3. RQ2. What Are the Attitudes Children from Smoking and Non-Smoking Households Have toward Physical Activity, Exercise, and Fitness?

#### 4.3.1. Awareness of the Physical Activity Guidelines

Twenty four percent of participants stated that children should participate in 60 min of PA per day. The current UK guideline for youth PA states children should do at least an average of 60 min MVPA per day [13]. Therefore, approximately three in four children were unaware of the current UK PA guidelines, which suggests that more promotion of the PA guidelines is required for children in this age group. Children from smoking households most often stated that children should do 90 min per day, whereas children from non-smoking households more frequently stated 60 min per day. Knowing how much PA children are recommended to participate in could be a potential facilitator for some children [89]. For girls aged 11–15, Roth and Stamatakis [89] found that knowing the PA guidelines was associated with meeting them, but the association was weak among boys. A Northern Ireland based study with adults found that 47% of respondents were unaware of the PA guidelines; males with lower education and more deprivation and females who are younger and in poor health were more likely to be unaware of the guidelines [90]. As the present study highlights the disparity in knowledge of the PA guidelines between children from smoking and non-smoking households, physical activity promotion strategies should aim to include awareness of PA guidelines for children in this age group.

#### 4.3.2. Is Fitness Important?

Fitness was important to all participating children but there were similarities and differences between the two exposure groups as to why fitness is important. The most common reason centred around capability, which was made up of physiological consequences, performance, and future capability, with the latter two being more important to children from the non-smoking household group. The theme of future capability encompassed children’s beliefs that being fit would benefit them as adults including in their careers: “because when I’m older I want to be an actress, and I want to be able to do a lot of stunts” (G2A/NS) and in family life: “So when you’re an adult and you have kids, and you take them out and make sure they’re fit as well” (G12C/NS). Children may be reflecting on what they have seen and heard from influential adults in their lives, including parents and teachers. The children are also showing evidence of ‘future thinking’ [91] and an element of delayed gratification [92,93] by acknowledging that fitness is also important for their future selves.

Children from non-smoking homes described health reasons for why fitness was important: ‘Because it’s good for your body and your bones and stuff,” (G8C/NS), but for children from smoking homes this was often centred around weight status. Perceptions that fitness is the absence of being overweight has been found in previous research with children [94], whose responses to what fitness means is mirrored by those of children in the present study: “Getting fit basically just means, like, non-fat”. Such differences may be due to echoing of parental attitudes but might also be reflecting children’s individual concerns and insecurities, as previous research found children from smoking homes were more likely to be overweight or obese [48].

Children from non-smoking homes also felt fitness was important to them for reasons relating to performance (in PA and sport), whereas children from smoking homes were more concerned with the physiological consequences of fitness. The variation may be explained by differing levels of fitness between the two groups. The children from smoking households have lower CRF [48] and were more likely to mention ‘getting out of breath’ than those from non-smoking households.

#### 4.3.3. How Can Children Improve Their Fitness?

There was consensus across both exposure groups about how they could improve their fitness, with slight differences in individual, social, and environmental factors [95]. Overall, children believed they could improve their fitness by increasing PA, through significant others, spending more time outdoors, improving their diet, and with the use of active equipment.

Children from both groups identified individual factors such as diet and weight for improving fitness, which suggests children are thinking about physical fitness rather than CRF only. A US study with similar age groups found that most children did not usually think about food choices [96]. Children have limited control over their own diet, as one participant suggests she can improve her diet with the assistance of a parent ‘I would say to my Mum, “I’m not having any like carbs or junk for maybe two months or something.’” (G8B/NS). Children identified that they should consume less ‘junk’ food and ‘fast food’ and more ‘healthy food’ instead. Although ‘healthy food’ was not defined in this discussion, their previous statements defined healthy food and drink as fruit, vegetables, and water. 

Children identified parents, friends, and dog ownership as social factors for improving fitness. Interestingly, only children from smoking homes identified dogs as a method of improving their fitness (taking their dogs for more regular walks), whereas only children from non-smoking homes identified parents. Yet the reverse finding is apparent when children were asked about facilitators for PA (see Section 4.4.1). Parents that value PA and fitness serve as role models, transmitting their desirable habits to their children [97] and parental exercise is positively associated with children’s sport participation and fitness [98]. Social factors are key determinants of PA in children, with participation with family and friends positively correlated with PA in children [99,100]. As social factors are important to these children, peer-group or family focused interventions may be an effective strategy for improving CRF.

For environmental factors, often the two ideas of more PA and spending time outdoors would go hand-in hand, for example, “Getting outside more and start being more active” (G9B/S). To improve their fitness, children’s recommendations are to be more physically active and spend more time outdoors. These suggestions are sensible as time spent outdoors is positively associated with MVPA [101,102,103] and MVPA is a determinant of CRF in children [2]. A systematic review by Hoyos-Quintero and García-Perdomo [104] concluded that environmental factors, such as playing in open spaces, have strong influences on children’s PA in early childhood, although the review focussed on a younger population than the present study.

The availability of equipment, for example, bicycles, scooters, treadmills, and trampolines, were often referred to by children from both smoking and non-smoking homes. According to Dumuid [105] possession of active play equipment, with the exception of bicycles, is not necessarily related to children’s MVPA. This is highlighted by a participant in the present study: “I think I should have my friends in more, because I have a big trampoline, but I usually don’t go on it unless I’ve got someone to go on it with, and that way I’ll be exercising and enjoying myself.” The gap between active equipment ownership and equipment use may explain inconsistencies in the literature regarding the relationship between ownership and MVPA [105,106]. Previous studies have found that children from low-income households, which includes most participants in the present study, had less access to active play equipment, such as bikes and jump ropes, [24] and to a garden or green space for outdoor play [107,108]. Interventions seeking to improve MVPA and CRF in this population should therefore consider strategies that support accessibility to outdoor spaces and the use of active equipment. Further research could explore access to outdoor space and active equipment for children from smoking households, which may, in turn, determine whether this group require specific interventions to improve their access.

### 4.4. RQ3. What Are the Perceived Barriers and Facilitators to a Child’s Ability to Be Physically Active and Does this Differ for Children from Smoking and Non-Smoking Homes?

#### 4.4.1. Perceived Barriers and Facilitators

As a number of factors were described as both facilitators and barriers of PA by the participants, such themes are discussed in conjunction below. For example, psychological factors were discussed positively as facilitators, such as ‘feeling motivated’, as well as negatively as barriers, e.g., ‘being in a bad mood’. Psychological status appeared to be an important facilitator and barrier to both groups of participants, which is a finding that is consistent with previous research in youth [109]. Intrinsic motivation for PA and sport has been found to be associated with PA, particularly in boys [110] and overweight and obesity is linked to less positive attitudes toward PA [111]. A study by Chen and Gu [112] has shown that adolescents with positive attitudes towards PA are more likely to be active and have higher CRF.

Participants believed that opportunities for PA, for example taking part in sports, facilitated their ability to be active. In a similar study in the US which utilised focus groups to explore perceived barriers and facilitators for PA, accessibility to PA was found to be a major barrier as parents and children voiced concerns that there was little access to PA opportunities [113]. Taking part in sports, whether as part of a club, at school, or unstructured with friends, was often discussed as an opportunity to facilitate PA for the participants in the present study. Coté et al. [114] highlights five psychosocial benefits conferred from sampling a range of sports during childhood: (1) life skills, (2) prosocial behaviour, (3) healthy identity, (4) diverse peer groups, and (5) social capital. Studies have also shown that childhood sport participation is an important correlate of PA in adulthood [60]. Additionally, organised sport participation has been found to be associated with increased fitness levels irrespective of enjoyment [115]. However, studies have demonstrated that financial barriers can restrict sport participation among children from low-SES [116,117] and participants in the present study, who are generally low-SES, identified finance as a barrier to PA participation. Children from low-income families spend less time in out-of-school structured activities, such as sport sessions, but may make up PA time in unstructured activity [118]. However, structured PA may confer additional benefits and increased MVPA [119,120]. Activity promoting voucher schemes may offer valuable assistance to children from low income smoking homes as this has been previously shown to overcome financial barriers to PA in children [121,122], conferring improvements in MVPA, fitness, and socialisation [123,124].

In addition to sports, participants often referred to unstructured opportunities for PA such as running, walking, cycling, and playing with friends. Brockman et al. [125] found that UK children from low-SES schools reported participating in more unstructured activities such as ‘free play’ with friends, whereas children from middle/high SES schools engaged in more sports clubs and organised activities. Structured activities require scheduling and time and time availability was a barrier identified only by children from non-smoking homes, who mentioned their commitments to other organised activities and sports clubs, not having enough time in the day, and having to wake up very early to get to morning training.

Intervention strategies to improve PA in children should therefore be population and context specific. As participants from non-smoking households appeared to have access to structured PA, interventions could focus on the provision of structured opportunities for PA to low-SES families, perhaps through the provision of sport participation vouchers as described above. 

The theme of the environment was made up of facilitators (transport and the outdoors) and barriers (transport and school). Children discussed the need for transport to get to places where they can participate in PA. Lack of transport was raised as barrier by one participant from a smoking household and is identified by low-SES groups in previous studies [117]. There is considerable research exploring transport, physical activity [126,127], and active travel [128]. Participants in the present study rarely discussed active travel, which may indicate a lack of awareness of opportunities to improve PA. Active transport methods could be promoted with this population, as a method to achieve 60 min of daily MVPA. Although children did comment that school and homework prevented them from being active, previous discussions indicated that children took part in PA at break, lunch, and after-schools clubs, but analysis of PA during the school day was not within the scope of this research. 

Although not discussed as a facilitator by children from smoking homes, a theme of outdoor play was developed from the responses of children from non-smoking households. Access to outdoor space is a correlate of PA [129] and increased outdoor time is associated with more minutes MVPA [130]. Parents of young children with lower educational backgrounds have been shown to use yards as a provision for PA more frequently than parents with higher educational backgrounds [131]. ‘Outdoor play’ was often discussed in conjunction with the opportunity for PA and significant others by participants. The ‘outdoors’ were also discussed by participants in relation to improving fitness and constituted a theme in relation to another research question that was discussed above.

Significant others, consisting of friends, adults, siblings, and dogs, were important facilitators for participating children, although dogs and siblings were only noted by children from non-smoking households and adults more frequently by children from smoking households. The provision of social support from significant others has been found to be a significant facilitator for children’s PA [113,132,133,134]. Social support from adults, in particular, is explored in more detail below (Section 4.4.2). According to Duncan et al. [135], children from low-income families perceive less sibling social support for PA compared to children from higher income families. Similarly, in the present study, only children from non-smoking households identified siblings as facilitators of PA, often in terms of co-participation. Dog ownership, identified as a PA facilitator by children from non-smoking households, is associated with higher levels of PA [136] and greater odds of meeting PA guidelines [137]. A qualitative study with a similar population to the present study also found that dog ownership was an enabling factor to PA [138].

Hohepa et al. [139] argued that social networks, including friends and school peers, need to be considered during the development of PA promotion strategies since low peer support was associated with reduced odds of activity in children. Friends and peer groups have great influence on behaviour [140] and friendship groups often have similar levels of PA [141]. In the present study, children from both smoking and non-smoking homes identified friends as facilitators of PA, acknowledging that playing with friends offers more enjoyment and provided support through co-participation. In previous research, friends have been shown to enhance enjoyment [54,142] and motivation for PA [143]. Therefore, PA promotion strategies for this population could target friendship groups as they may have similar activity levels and will offer peer support.

Screen time was highlighted as a significant barrier by children from both smoking and non-smoking households. Technology has been found to be a perceived barrier to children’s PA in other qualitative studies [113,144,145] and smartphone and tablet use has been shown to be associated with lower PA in adolescents [146]. Children from smoking households mentioned sedentary behaviours (other than screen time), such as ‘lying on the couch’, as barriers to their PA more frequently than children from non-smoking households, who more frequently discussed screen time in particular. Tandon et al. [24] have shown that lower SES home environments provide more opportunities for sedentary behaviour and fewer for PA, which is relevant to the present study as the majority were from low-SES. Participants from smoking households classified one sedentary behaviour, ‘sleep’, as a facilitator of their PA, explaining that getting enough sleep allows them to be more active. This factor linked closely with the psychological factors identified by these children such as ‘feeling too tired’. Interestingly, all references to feeling ‘too tired’ as a barrier to PA were made by participants from smoking households.

Physiological factors, such as health, diet, and sleep, were discussed as both barriers and facilitators to PA. Diet was discussed far more frequently by participants from non-smoking homes than those from smoking homes. One participant identified chocolate as both a facilitator and a barrier as “chocolate will give you a sugar rush if you eat too much” (B10C/NS), indicating a consideration for the nutritional cost and benefits of food. Children from non-smoking households appeared to be very aware of the need to eat ‘healthy’ food and less ‘junk food’ and ‘sweets’. O’Dea [79] used focus groups with child participants, who also identified ‘junk food’ as a barrier to PA due to the ‘sluggish’ feeling associated with eating such foods. As most children from smoking households did not discuss diet, it may be that diet is not a perceived as a facilitator or barrier for these participants, or that hey were thinking about more direct influences on PA rather than indirect factors such as diet. Health (including injury) was discussed as a facilitator and barrier more often by children from non-smoking households. Injuries were often discussed as important barriers to PA, along with references to physiology including the heart: “If you have a heart. It pumps your blood round.” (B12C/S). The finding that health, including ‘a healthy body’, is discussed more by children from non-smoking households may further indicate a greater level of health literacy [81] or physical literacy [82] in this group, as described above in relation to the reasons why children take part in PA.

#### 4.4.2. How Do Adults Limit or Facilitate Children’s Physical Activity According to Children from Smoking and Non-Smoking Households?

Children expressed that adult support was provided in logistical and financial forms, opportunities for PA, and through verbal instruction and encouragement. Restriction from adults was due to punishment (grounding) and safety concerns. The findings of the present study are consistent with Noonan et al. [54], who also found that whilst logistical forms of support are correlates with child PA, they are less influential than verbal methods of support such as encouragement.

Children from non-smoking homes discussed verbal encouragement from adults as a method in which adults facilitated their PA, referring to adults as ‘motivating’ and that adults encourage the children to ‘do their best’. Although children from smoking homes did report verbal *instruction* from adults as a facilitator for PA, for example ‘get off your computer and go play outside’, children from smoking homes did not refer to verbal encouragement. Noonan et al. [54] found that verbal encouragement had the greatest effect on children’s emotions and their PA, although ‘encouragement’ also included ‘instruction’. In the present study, encouragement and instruction have been separated to show the nuances of the language used. Through the words of encouragement and instruction echoed by the children, it seems apparent that both parents who smoke and do not smoke are aware that children should be physically active and try to facilitate this through verbal methods. Brockman et al. [125] found that children from high-SES schools were assisted in PA through actions such as logistical and financial support, whereas children from low-SES schools were encouraged through verbal encouragement and demands. Hohepa et al. [139] found that children that receive high levels of encouragement from parents were more active, regardless of whether encouragement was provided by two parents or the sole parent. The findings in the present study for low-SES homes are consistent with those in Brockman et al. [112], as much of the facilitation for children’s PA by adults was through verbal instruction.

Children from smoking homes identified parental provision of opportunities for PA more than children from non-smoking homes. They talked about parents facilitating their PA by going to the park as a family, walking to the shops or school, and cycling together. Participating parents in a study by Joseph et al. [144] acknowledged that taking their child to specific locations, such as the park, could help facilitate more PA and parents have previously expressed the desire for more opportunities for parental involvement [145]. Parental PA and sport participation influences offspring PA [147,148] and MVPA [149]. A longitudinal study has shown parental PA to be associated with offspring PA from childhood until middle age [150]. Parental modelling and support also influences child and adolescent PA [151,152]. It is clear from the focus group discussions, particularly of those from smoking households, that children perceive parents to be aiding their PA levels by proving opportunities including family co-participation in PA. However, an apparent difference between both groups of children is the type of PA provided by parents, with children from non-smoking homes more frequently discussing structured PA. Structured PA, such as organised sport, may result in higher levels of MVPA [119,120] and increased levels of fitness [115] compared to unstructured PA. However, time spent outdoors, whether structured or unstructured, results in more active time and MVPA than time spent indoors [120]. The benefits of structured PA participation are discussed above in relation the facilitators and barriers to PA and include enhanced MVPA and increased fitness, as well as many psychosocial benefits. Some children commented how ‘inspiring people’ and role models often motivate them to be engaged in PA. Recent research has shown that family-based interventions are rated as more ‘fun’ and result in greater improvements in MVPA [153] and so strategies to enhance PA could therefore target co-participation via family-wide interventions, which would confer the additional benefits of social support and adult behavioural modelling. Social support from family, friends, teachers, and coaches could also be utilised in strategies to assist children in overcoming the discussed perceived barriers to PA for this population.

### 4.5. RQ4. What Are Children’s Perceptions of Their Own Fitness and Physical Ability and Does This Differ for Children from Smoking and Non-Smoking Homes?

No difference could be observed between the perceived fitness scores for children from smoking and non-smoking households. Most children perceived their own fitness above average (e.g., more than 5 out of 10). This finding may be explained by the better-than-average-effect; the tendency to evaluate oneself more favourably than an average peer [154]. Some studies have shown that self-perception is strongly related to physical fitness and motor competence [155,156,157], whereas others have found only moderate correlation [73] or no correlation [158]. Most children in the present study estimated their fitness very highly. Weiss and Amorose [159] also found that similarly aged children had higher than actual self-perceptions of motor competence. Previous studies involving youth have found participants to overestimate their motor competence [87] and movement skill competency [158]. In terms of participation in PA, overestimation is preferred as underestimation may negatively influence motivation and greater self-perception increases participation in PA [158]. It is therefore a positive finding that children in the present study, from smoking and non-smoking households, have inflated perceptions of their own fitness, as this is likely to encourage motivation for and participation in PA. However, as self-perception accuracy increases with age [155,159], if the children with low fitness remain low-fitness into adolescence, then their self-perception may decrease accordingly. Social desirability bias may have impacted the children’s choices of self-perception scores by choosing to either increase or decrease their scores based on another child’s response and due to the influence of social norms [64]. A strategy to reduce peer influence on participants’ rating of their own fitness would be to have children rate their fitness individually and privately, for example, via questionnaire.

Boys and girls from smoking homes both rated running as the hardest activity during the pictorial task, for reasons such as not enjoying it and because it is ‘tiring’. For children from non-smoking homes, gymnastics (boys) and monkey bars (girls) were rated as the hardest, followed by running. Children from smoking households generally had lower CRF in this sample and so may genuinely experience running as physiologically more difficult. Children’s perceptions of difficulty suggest that aspects of fitness other than CRF, regardless of the metabolic demand (METs) [160] of the activity, are used by children to determine how ‘difficult’ or ‘hard’ an activity is. For example, the ‘monkey bars’ require the component of fitness that is strength, whereas the ‘crab’ requires flexibility, agility, and strength. Participants that rated activities other than running as the most difficult may have low perceived competence in particular aspects of fitness, such as strength and flexibility, compared to CRF. The sex differences in perceived difficulty of the monkey bars and gymnastics demonstrated could also be subject to group desirability bias.

The present study was conducted prior to the COVID-19 pandemic and before national lockdowns were implemented in the UK. Since early 2020 however, COVID-19 restrictions have resulted in children spending more time at home and indoors and physical activity levels have decreased [161]. Such is concerning not only for children’s PA, but also for children from smoking households who may be exposed to much greater levels of second-hand smoke during the pandemic due to the requirement to stay at home and stay indoors. For children from smoking homes, the hours spent at school and extra curriculars were hours free from SHS, but spending all day at home greatly increases SHS exposure. COVID-19 restrictions may have therefore exacerbated the negative impact of SHS on children’s health and fitness. It is probable that due to the decrease in PA and increase SHS exposure, children from smoking homes will have a decline in CRF due to the pandemic and the associated restrictions.

### 4.6. Strengths and Limitations

The above findings should be interpreted in light of a number of limitations. Although the sample size was relatively small with 38 participants, the small sample permitted the generation of rich data, which is a major strength of the study. Future research could expand on this sample population to include participants from other regions of the UK, as well as participants from a wider age range. Younger children and adolescents may have different thoughts and perceptions surrounding PA and fitness and face different barriers and facilitators. The majority of participants lived in neighbourhoods within the lowest two deciles for deprivation based on the English Indices of Multiple Deprivation (EIMD). However, low-SES areas were targeted in order to recruit as many tobacco smoking families as possible. The findings are therefore less applicable to children of medium-high socioeconomic status. The sample population was diverse and represented a range of ethnicities and backgrounds, with approximately 34% of the sample that were made up of ethnicities other than White British. However, only 26.5% of families invited to participate in the research gave permission for their children to do so, which is likely due to the combination of the contentious nature of the work and the use of the opt-in approach to recruitment. Further research on the topic may benefit from an opt-out approach.

The use of self-reported smoking status by parents is a limitation of this study and it is recognised that surveys are potentially subject to biases, including desirability and recall bias. Cotinine analysis could not be used due to financial restrictions and concerns for participant burden, but eCO was used as a measure of acute SHS exposure in children. However, eCO concentrations were not significantly different between groups. Future work would benefit from the use of salivary cotinine as a biochemical indicator for recent SHS exposure in addition to self-report surveys.

Due to ethical concerns regarding eliciting anxiety within the children when considering the smoking status of their parents and family members, tobacco smoking was not discussed with the children. As a result, we were not able to gain insight into children’s opinions and thoughts about smoking or second-hand smoking or how having a smoking family member made them feel. This information would be highly valuable and could aid campaigns to prevent smoking uptake as well as smoking cessation. One question in particular, which required children to rate their own fitness level, would have been more suitably posed individually rather than in a group setting where children may be influenced by their peers. In addition, whilst the focus group has strengths in eliciting group discussion, there were some children that were considerably less talkative than other children. Some children may not have felt comfortable sharing thoughts with the group. We attempted to mitigate this barrier by using more interactive methods and this is discussed as a strength below.

A major strength of this research is that, to the authors’ knowledge, it is the first to represent the thoughts, beliefs, and perceptions surrounding physical activity and fitness relative to children from smoking households. This research provides a voice to a population of children who may have additional health risks due to second-hand smoke exposure and may face other barriers to PA and fitness than their non-smoking household counterparts. A second strength is the use of activities in addition to discussion. Using sticky notes, writing, photographs, and diagrams aided discussion and allowed shy children to communicate in alternative methods. The research aimed to elicit as much information from children from smoking households as possible, through a multitude of methods. A further strength of the work is that it is informed by a complimentary quantitative study. Analysis was able to utilise information gathered from quantitative aspects, such as fitness scores, which allowed for better informed analysis. 

## 5. Conclusions

This study used focus groups to explore the thoughts, opinions, perceptions, and beliefs surrounding physical activity and fitness of children from non-smoking and smoking households.

The findings support the main hypothesised mediators of PA in children including self-efficacy, enjoyment, perceived benefit, and social support. However, the variations in children from smoking and non-smoking households’ reasons for taking part in PA indicate the need for targeted interventions. Strategies to increase participation in PA for children from smoking households could focus on facilitating friendship/peer group physical activities that they regard as ‘fun’ and ‘enjoyable’.

As less than a quarter of participants were aware of the PA guidelines, strategies to improve children’s awareness of these guidelines are recommended to increase PA participation. Whilst all children agreed fitness was important to them, differences emerged between groups for why. Interventions to improve CRF in this population should support access to PA participation by providing active equipment and safe outdoor space.

Perceived barriers and facilitators are similar to previous research, but variances emerged for important barriers and facilitators in children from smoking and non-smoking homes. Strategies to overcome barriers should be based on household smoking status and focus on using perceived facilitators. The majority of children perceived their own fitness to be high or above average. Variances were observed in ranking physical activities by difficulty between boys and girls and exposure group, with children from smoking households rating running as the hardest.

A handful of themes overlap several research questions and thread through children’s responses throughout the study. Examples include health, significant others, opportunity for physical activity, and the outdoors. Significant others (family, friends, and dog ownership) are an important theme for participants from both smoking and non-smoking households concerning why they participate in PA (RQ1), methods to improve fitness (RQ2), and a key facilitator of PA (RQ3). Health was a theme developed in response to children taking part in PA for health benefits (RQ1), why fitness is important to the participants (RQ2), and as a perceived barrier/facilitator to PA (RQ3). The outdoors was a theme relevant to how children can improve their fitness (RQ2) and as a facilitator for PA (RQ3). Diet was discussed as a method to improve fitness (RQ2) and as both a barrier and facilitator to PA (RQ3). These themes are important to participants and should be considered when planning PA and CRF interventions.

The difference in perceptions of the health benefits of PA and fitness between children from smoking and non-smoking households warrants further exploration. Throughout the study, participants from non-smoking households demonstrated greater awareness of the PA guidelines, referred to extrinsic motivators of PA, referred to the health benefits of fitness, and had considerations for the future self. Future work should compare physical literacy, in particular the psycho-social/cognitive factors, of children from smoking and non-smoking households.

To the authors’ knowledge, this important work is the first to explore perceptions of children from smoking and non-smoking households regarding physical activity and fitness. Interventions to improve the levels of PA and CRF in children from low-SES and smoking households could benefit from these child participant’s perspectives in order to create relevant and effective strategies.

## Figures and Tables

**Figure 1 children-08-00552-f001:**
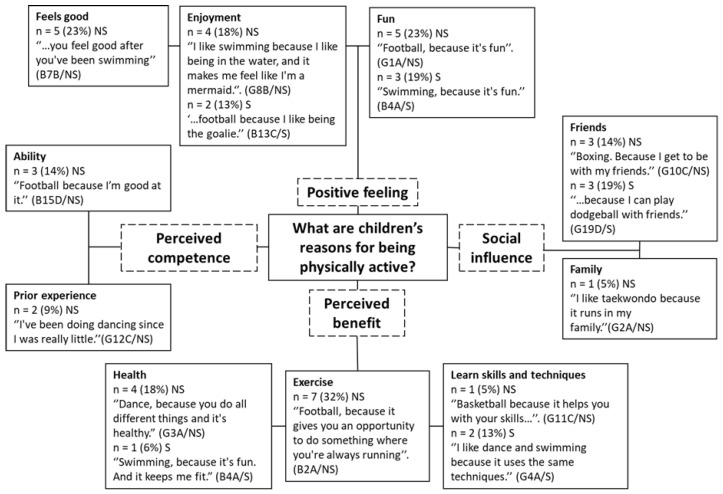
Pen profile demonstrating children’s reasons for being physically active for children from non-smoking (NS) homes and smoking (S) homes. Percentages represent the proportion of each group that contributed to the theme for children from smoking (*n* = 16) and non-smoking homes (*n* = 22).

**Figure 2 children-08-00552-f002:**
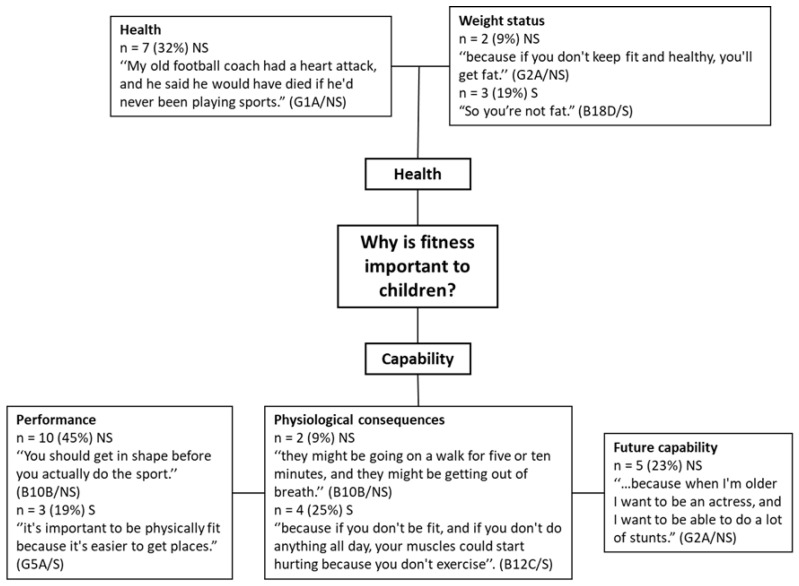
Pen profile showing children’s reasons why fitness is important to them. S = children from smoking households, NS = children non-smoking households. Percentages represent the proportion of each group that contributed to the theme for children from smoking (*n* = 16) and non-smoking homes (*n* = 22).

**Figure 3 children-08-00552-f003:**
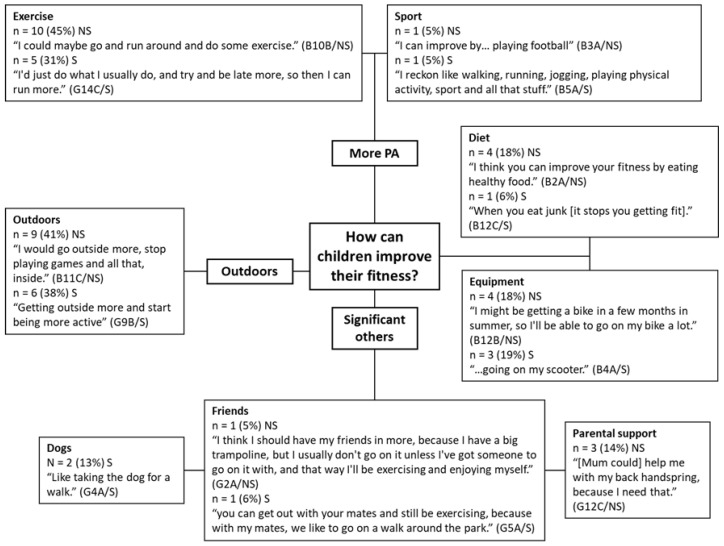
How can children improve their fitness according to children from smoking (S) and non-smoking (NS) homes. Percentages represent the proportion of each group that contributed to the theme for children from smoking (*n* = 16) and non-smoking homes (*n* = 22).

**Figure 4 children-08-00552-f004:**
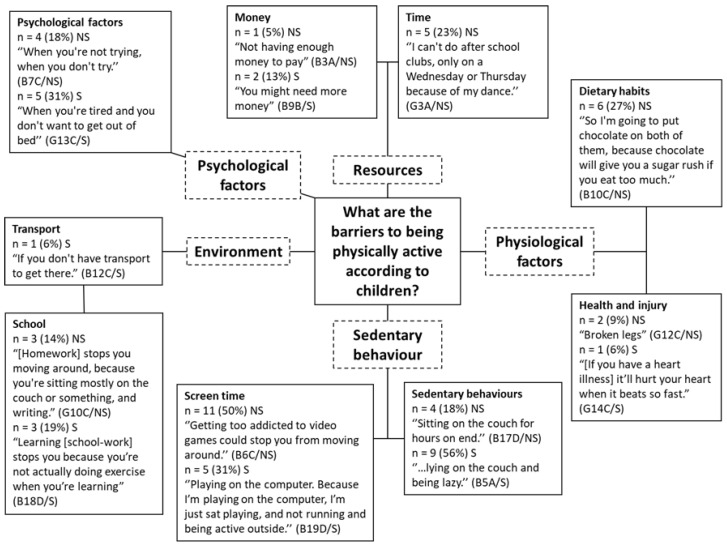
Pen profile demonstrating the barriers to being physically active according to children from non-smoking (NS) and smoking (S) homes. Percentages represent the proportion of each group that contributed to the theme for children from smoking (*n* = 16) and non-smoking homes (*n* = 22).

**Figure 5 children-08-00552-f005:**
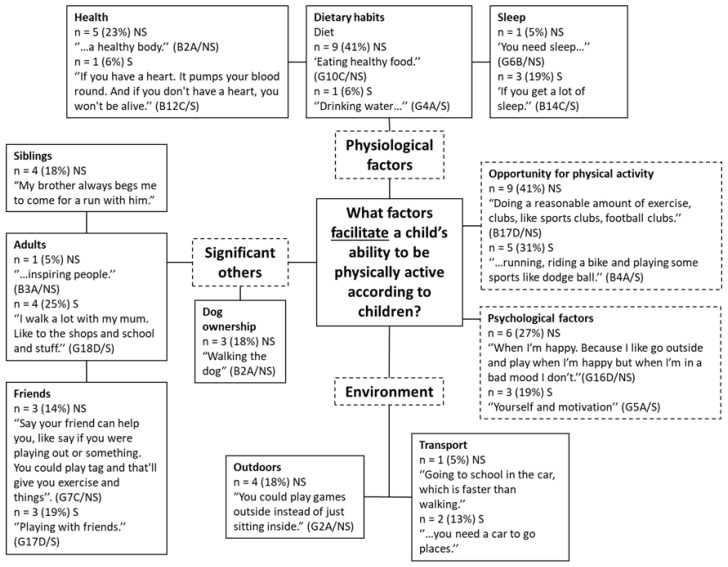
Pen profile demonstrating what factors facilitate a child’s ability to be physically active according to children from non-smoking (NS) and smoking (S) homes. Percentages represent the proportion of each group that contributed to the theme for children from smoking (*n* = 16) and non-smoking homes (*n* = 22).

**Figure 6 children-08-00552-f006:**
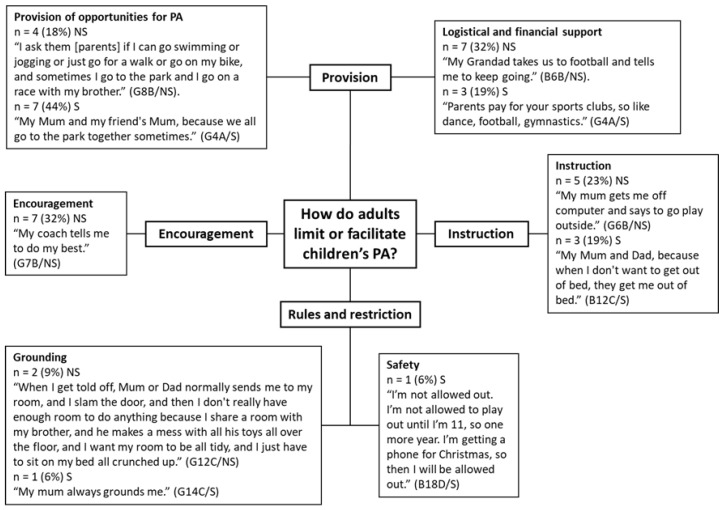
Pen profile demonstrating how adults limit or facilitate a child’s ability to be physically active according to children from non-smoking (NS) and smoking (S) homes. Percentages represent the proportion of each group that contributed to the theme for children from smoking (*n* = 16) and non-smoking homes (*n* = 22).

**Table 1 children-08-00552-t001:** Focus group membership.

Focus Group	School	Girls	Boys	Non-Smoking Household	Smoking Household
1	A	2	3	5	0
2	A	3	2	1	4
3	B	2	2	4	0
4	B	2	3	3	2
5	C	3	1	4	0
6	C	2	3	0	5
7	D	2	3	5	0
8	D	3	2	0	5
Total		19	19	22	16

**Table 2 children-08-00552-t002:** Activities ranked in overall order of ‘hardest’ to ‘easiest’ as described by participating children by household smoking status and sex.

Perceived Difficulty	Non-Smoking Boys	Smoking Boys	Non-Smoking Girls	Smoking Girls
**Hardest**	Gymnastics	Running	Monkey bars	Running
	Running	Gymnastics	Running	Gymnastics
	Swimming	Swimming	Swimming	Monkey bars
	Monkey bars	Monkey bars	Gymnastics	Swimming
**Easiest**	Walking	Walking	Walking	Walking

## Data Availability

The data presented in this study are available upon reasonable request from the corresponding author. The data are not publicly available due to privacy and ethical considerations.

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
