# Peer review of "Children of Smoking and Non-Smoking Households’ Perceptions of Physical Activity, Cardiorespiratory Fitness, and Exercise"

_children, 2021, doi:10.3390/children8070552_

Round 1

Reviewer 1 Report

This is an important paper providing evidence to support tobacco control. It is surprising that no reference is made to the Framework on Tobacco Control or MPOWER or to Cochrane reviews Legislative smoking bans to reduce Secondhand smoke exposure 2016 or Family and carer smoking control programmes for reducing children's exposure to environmental tobacco smoke 2018.

The methods section provides no details confirming the status of smoking or non-smoking homes - this is critical re analysis has been presented and title of the paper.  How did parents confirm household status ? A self-report question or via biochemical measure. This information should be included and, if self-reported, then add to limitations. 

If the confirmation of smoking status is self-report - we are aware of underreporting of smoking status by up to 40%. In addition, were parents asked if they used e-cigarettes / vaping.

The impact on health is noted - however, are there other variables that could explain the outcomes? Is it only smoking status that is associated with the outcomes. Had children medical histories that contributed or were children confirmed they to be clear of any underlying conditions, including asthma.

Not all children who attend schools in deprived areas are deprived.  Are there data on ethnicity or country of origin. Many families who migrant to a country may only have access to specific schools due to entrance policies in others.  Recruitment required reading skills and for some parents this may have been a barrier to participation from the outset -- was any attempt made to understand why so few engaged with the study? 

How was SES measured in this study?

Including accounts from children is important. Can I clarify that children were not asked about the smoking status of parents - so the entire study sought information on physical activity. Did children self report their fitness level or was it measured?  How did results compare to other studies that have asked 9 to 11 years about their fitness on a scale of 1 to 10?  Did children hear each other's scores during a focus group or how was this data captured. Could children have influenced each other's scores of perceived fitness for example?

Smoking status is an independent variable and does not appear to be biochemically confirmed.

Author Response

We would like to thank the reviewers for their thoughts and suggestions within this first round of review. Within this revision, changes have been made as requested throughout the manuscript. Primarily, a descriptive section has been added regarding how household smoking status was established, and the main limitations such as self-report survey data, and low response rate, have been addressed.

We hope that these additions and amendments address the reviewers’ thoughts and suggestions. We provide a detailed point by point response (attached) and have highlighted the amendments within the submitted manuscript in yellow.

Reviewer 2 Report

This manuscript describes in great detail a study of children from smoking and nonsmoking home environments and their perception of physical activity.  The authors are thorough, but the steady reporting of percentages and ns belies the fact that the data come from small and potentially quite biased samples; there is no need to placate quantitative colleagues with extensive numbers when the often subtle qualitative differences between the two main groups are not statistically significant.  

Minor suggestions: 

p. 1 line 20 Fewer than instead of Less than.

p. 1 lines 39-40 Thus...region is a sentence fragment.  

p. 2 69 of which should perhaps be on which (=which there is less research on, not of)

p. 3. The authors describe study participants in some detail but do not say why they targeted 9-11 year-olds.

p. 3 participants were from only 4 schools of many approached, and 26.5% had the parental permission to proceed.  Thus the sample appears to have been biased from the start.

p. 5 section 2.5 Migration of data is a strengh of the paper, demonstrating how data were properly classified in smoking and non0-smoking home environments.

p. 5 lines 231-235.  The authors mention pen profiles and provide a couple of references, but they never summarize in a sentence or two what exactly pen profiles are.  Since, as the paper proceeds, the diagrams are labeled as pen profiles, readers can figure it out if they were previously unaware of the term.  The diagrams/pen profiles convey information succinctly, rendering some of the text superfluous or repetitive.

p. 11 lines 437-439: As noted above, the detail of listed themes is admirable, but this reviewer questions the value of representing such preciion when the sample seems so biased and the numbers are relatively small. Is this a case of qualtative methodologists having to prove themselves to quantitatively oriented colleagues? 

p. 23 lines 1003-1009.  Citing ethical concerns (asking about parental smoking might make the children feel bad?), the authors did not ask directly about the central set of relationships in the study: whether or not a home environment had second-hand smoke, and how that might influence the youths' physical activity and their attitudes towardd PA.  This was a missed opportunity.

Small typo on p. 24, line 1066: author's should be authors' since they are multiple coauthors.

[In the discussion, although entirely optional, might the authors be able to relect upon what their children would have experienced in the context of the pandemic, including greater exposure to secondhand smoke if the family smoked in the home (vs. school), and how that might influence views and activity in the domains of physical activity?]

Many of the aforementioned points shoul be able to be dealt with fairly easily.  

Author Response

(The authors gave the same response as above.)

Round 2

Reviewer 1 Report

Thank you for providing detailed feedback and amending manuscript.

Author Response

We would like to thank the reviewer and editor for their suggestions within the second round of review. 
The reference has been integrated within the manuscript at lines 73-75 and reads '...and there is consistent evidence that national smoking bans have improved cardiovascular health outcomes, and reduced mortality for associated smoking-related diseases [34].'
We hope this addition addresses the reviewers' suggestion.